# Universal Successor Features for Transfer Reinforcement Learning

## Abstract

Transfer in Reinforcement Learning (RL) refers to the idea of applying knowledge gained from previous tasks to solving related tasks. Learning a universal value function (Schaul et al., 2015), which generalizes over goals and states, has previously been shown to be useful for transfer. However, successor features are believed to be more suitable than values for transfer (Dayan, 1993; Barreto et al., 2017), even though they cannot directly generalize to new goals. In this paper, we propose (1) Universal Successor Features (USFs) to capture the underlying dynamics of the environment while allowing generalization to unseen goals and (2) a flexible end-to-end model of USFs that can be trained by interacting with the environment. We show that learning USFs is compatible with any RL algorithm that learns state values using a temporal difference method. Our experiments in a simple gridworld and with two MuJoCo environments show that USFs can greatly accelerate training when learning multiple tasks and can effectively transfer knowledge to new tasks.

## 1 Introduction

Deep Reinforcement Learning (RL) has been able to achieve human-level performance in many domains, such as playing Atari games (Mnih et al., 2015) and controlling robotic arms (Levine et al., 2016). However, these methods often spend a huge amount of time and resources to train a deep model to perform a single task. Currently applying knowledge gained from performing a single task to performing another related task remains a challenging problem. This is the problem Transfer learning is trying to solve (Taylor & Stone, 2009). Evidently though transferring knowledge from one task to another would not be possible if the two tasks were completely unrelated. Therefore, in this work, we focus on one particular transfer scenario, where dynamics among tasks remain the same while their goals are different. We elaborate on this in Sec. 2.

What can we use as knowledge for transfer in RL? One form of model-free knowledge is the goal-specific state-values as in general value functions (Sutton et al., 2011). However, the state-values' dependency on the goal and its corresponding reward structure may be an obstacle when attempting to use these values for transfer. An alternative form of model-based knowledge is to learn the transition dynamics[1], which are independent of the reward function. However, when performing a rollout with the estimated transition dynamics prediction error will accumulate over time, which may limit the utility of this approach in practice. A third form of knowledge that may be suitable for transfer is successor features (SFs) (Dayan, 1993; Barreto et al., 2017; 2018), which can recover state-values without explicitly performing a rollout. Briefly speaking, SFs provide an action-contingent summary of the future. This third approach, cannot directly generalize to unseen new goals.

In this paper, we propose Universal Successor Features (USFs). Similar to how a universal value function (Schaul et al., 2015) learn to generalize values over goals, USFs learn to generalize successor features over goals. Since SFs for each goal capture certain knowledge about the underlying dynamics of the environment, USFs could learn the shared dynamics and exploit their generality over the goals. Furthermore, USFs are also able to recover values without explicitly performing a rollout. USFs result from decomposing universal state-values into USFs and reward features. Note that while we focus on state-action-values (or simply action-values) in this paper in order to tackle

---

[1] the action-contingent probability of transitioning from one state to another

control problems, this idea can apply to state-values as well. Unlike in Schaul et al. (2015), we learn these two decomposed factors jointly by exploring the environment while bootstrapping.

Our contributions are three-fold. First, we propose USFs and show how they can be applied to *any* RL algorithm that learns state-values using a temporal difference method. Second, we propose a flexible end-to-end architecture that is suitable to learn USFs by interacting with the environment. Third, our experiments demonstrate that USFs can greatly accelerate training when learning multiple tasks and can effectively transfer knowledge to new tasks.

This paper is organized as follows. In Sec. 2, we introduce USFs together with our end-to-end architecture, training procedure, and we further explain how USFs can be used for transfer Reinforcement Learning. Then, in Sec. 3, we demonstrate the results of applying USFs to a simple gridworld environment and two MoJuCo environments. Afterward, in Sec. 4, we discuss some of the most relevant papers in the literature. Finally we conclusions in Sec. 5.

## 2 UNIVERSAL SUCCESSOR FEATURES

Consider a Markov decision process (MDP) with state space $\mathcal{S}$, action space $\mathcal{A}$, and transition function $p$ where $p(s'|s,a)$ is the probability[2] of transitioning to $s' \in \mathcal{S}$ when action $a \in \mathcal{A}$ is taken in state $s \in \mathcal{S}$. We borrow the extension of the MDP framework to goals and state-dependent discounting from Schaul et al. (2015). For each goal $g \in \mathcal{G}$ (often $\mathcal{G} \subseteq \mathcal{S}$ but this is not an explicit requirement of this framework), define the pseudo-reward function $r_g(s,a,s')$ and pseudo-discount function $\gamma_g(s) \in [0,1]$. To define an episodic problem in this framework we can define $\gamma_g(s)$ to be 0 if $s$ is a terminal state with respect to $g$.

For any policy $\pi : \mathcal{S} \mapsto \mathcal{A}$ and fixed goal $g$, the *action-value function* is defined as

$$Q_g^\pi(s,a) = \mathbb{E}^\pi \left[ \sum_{t=0}^\infty r_g(S_t, A_t, S_{t+1}) \prod_{k=1}^t \gamma_g(S_k) \middle| S_0 = s, A_0 = a \right].$$

As we now show, we can factorize $Q_g^\pi$ into successor features and goal-specific features.

### 2.1 TRANSFER VIA UNIVERSAL SUCCESSOR FEATURES

As with Kulkarni et al. (2016) and Barreto et al. (2017), we assume that the reward function $r_g(s_t, a_t, s_{t+1})$ can be factorized as

$$r_g(s_t, a_t, s_{t+1}) = \phi(s_t, a_t, s_{t+1})^\top \mathbf{w}_g, \tag{1}$$

where $\phi : \mathcal{S} \times \mathcal{A} \times \mathcal{S} \mapsto \mathbb{R}^d$ represents the transitions and $\mathbf{w}_g \in \mathbb{R}^d$ are goal-specific features of the reward. Note that in this factorization only $\mathbf{w}_g$ depends on the goal. In many situations $\phi(s_t, a_t, s_{t+1})$ can be rewritten as $\phi(s_{t+1})$. To see why, consider a simple gridworld with $d$ cells. Here $\phi(s_t, a_t, s_{t+1}) = \phi(s_{t+1})$ can be a one-hot vector indicating the position of the agent at time $t+1$, while $\mathbf{w}_g$ can be a $d$-dimensional vector whose $i$-th entry is the immediate reward of reaching the cell $i$ under the goal $g$. If $\phi(s_t, a_t, s_{t+1})$ and $\mathbf{w}_g$ are defined in this way then whatever the previous state $s_t$ and action $a_t$ are, as long as the agent ends up in $s_{t+1}$, the transition is represented by $\phi(s_{t+1})$ and the agent will receive the corresponding reward from $\mathbf{w}_g$ for reaching that state. In other cases where the reward depends on the whole transition, we can easily modify subsequent discussion and model architecture to accommodate this.

Using the factorization in Eq. (1), the action-value can be rewritten as

$$Q_g^\pi(s,a) = \mathbb{E}^\pi \left[ \sum_{t=0}^\infty \phi(S_t, A_t, S_{t+1}) \prod_{k=1}^t \gamma_g(S_k) \middle| S_0 = s, A_0 = a \right]^\top \mathbf{w}_g = \boldsymbol{\psi}_g^\pi(s,a)^\top \mathbf{w}_g$$

where $\boldsymbol{\psi}_g^\pi(s,a)$ are the *Universal Successor Features* (USFs) of $s$ and $a$ for goal $g$. In the above gridworld example, $\boldsymbol{\psi}_g^\pi(s,a)$ would be a $d$-dimensional vector of discounted expected future state visitations that results after taking action $a$ in state $s$ and thereafter following policy $\pi$.

---

[2]or probability density in case of continuous state space

If $\mathbf{w}_g$ is given, then the following Bellman equations enable us to learn USFs the same way as we would learn the action-values:

$$Q_g^\pi(s, a) = \mathbb{E}^\pi[r_g(s, a, S') + \gamma_g(S')Q_g^\pi(S', A')], \quad \boldsymbol{\psi}_g^\pi(s, a) = \mathbb{E}^\pi[\boldsymbol{\phi}(s, a, S') + \gamma_g(S')\boldsymbol{\psi}_g^\pi(S', A')].$$

We can approximate $\boldsymbol{\psi}^\pi(s, a)$ using a model $\boldsymbol{\psi}^\pi(s, a; \boldsymbol{\theta}_\psi)$ parameterized by $\boldsymbol{\theta}_\psi$, so that our method can be used with any algorithm that learns action-values using a temporal difference method[3], including on-policy methods like SARSA (Sutton, 1996) and A2C (Mnih et al., 2016) as well as off-policy methods like DQN (Mnih et al., 2015) and DDPG (Lillicrap et al., 2016). One remaining problem is that we need to have $\mathbf{w}_g$ for a given $g$. One way to accomplish this is to model $\mathbf{w}_g$ as $\mathbf{w}(g; \boldsymbol{\theta}_w)$ and learn the parameters $\boldsymbol{\theta}_w$.

In order to utilize USFs for transfer, we can learn the parameters $\boldsymbol{\theta} = [\boldsymbol{\theta}_\psi, \boldsymbol{\theta}_w]$ from some "good" policies and their SFs for a subset of goals, then let the model generalize these learned SFs to the SFs of "good" policies of unseen goals. In the next section, we describe how one could train $\boldsymbol{\theta}$.

## 2.2 Training USFs

Suppose that at time $t$, we obtain a transition[4] $\{g, s_t, a_t, s_{t+1}, r_{t+1}, \gamma_{t+1}\}$ where $r_{t+1} = r_g(s_t, a_t, s_{t+1})$ and $\gamma_{t+1} = \gamma_g(s_t, a_t, s_{t+1})$. This transition can either come directly from the environment or may be sampled from a replay buffer. Recall that USFs can be applied to any base RL algorithm as long as it involves learning action-values using a temporal difference method. In the case of DQN, which naturally extends to our multi-goal setting (see Appendix A for more details), we update the parameters $\boldsymbol{\theta}_Q$ of the action-value function using (semi-)gradient descent on

$$[Q(s_t, a_t, g; \boldsymbol{\theta}_Q) - y_t]^2 \quad \text{where} \quad y_t = r_{t+1} + \gamma_{t+1}\max_a Q(s_{t+1}, a, g; \boldsymbol{\theta}_Q) \tag{2}$$

is considered a *fixed* target action-value (i.e., it is treated as a constant rather than a function of $\boldsymbol{\theta}_Q$). Similarly, to learn USFs' parameters $\boldsymbol{\theta}$ for *optimal* action-values[5], the next-state action $a^*$ could be

$$a^* = \operatorname*{argmax}_a \boldsymbol{\psi}(s_{t+1}, a, g; \boldsymbol{\theta}_\psi)^\top \mathbf{w}(g; \boldsymbol{\theta}_w).$$

Using this, we define the loss on $Q$ to be

$$L_Q = \left[\widehat{Q}_t - \boldsymbol{\psi}(s_t, a_t, g; \boldsymbol{\theta}_\psi)^\top \mathbf{w}(g; \boldsymbol{\theta}_w)\right]^2 \text{where } \widehat{Q}_t = r_{t+1} + \gamma_{t+1}\boldsymbol{\psi}(s_{t+1}, a^*, g; \boldsymbol{\theta}_\psi)^\top \mathbf{w}(g; \boldsymbol{\theta}_w) \tag{3}$$

is the *fixed* target action-value. Similarly, we define the loss on SF to be

$$L_\psi = \left\|\widehat{\boldsymbol{\psi}}_t - \boldsymbol{\psi}(s_t, a_t, g; \boldsymbol{\theta}_\psi)\right\|_2^2 \quad \text{where} \quad \widehat{\boldsymbol{\psi}}_t = \boldsymbol{\phi}(s_t, a_t, s_{t+1}) + \gamma_{t+1}\boldsymbol{\psi}(s_{t+1}, a^*, g; \boldsymbol{\theta}_\psi) \tag{4}$$

is considered the *fixed* target SF. This loss ensures that the learned SF will satisfy its own Bellman equation. As we will show in the experiments, this loss is resilient to the reward structure of the MDP. Although we can separately learn $\mathbf{w}$ based on a regression loss from Eq.(1), it is less informative in terms of guiding the agent compared to regression based on Eq.(3), the actual quantity for decision making (results in Appendix D). Our final loss function is $L = L_Q + \lambda \cdot L_\psi$ where $\lambda > 0$ is a hyperparameter. With this loss, we can perform updates based on the (semi-)gradient $\nabla_{\boldsymbol{\theta}} L$.

To see how we can use USFs for transfer, note that because learning is based on the optimal action, Eq.(3) and (4) learn the $\boldsymbol{\psi}$ that corresponds to the *optimal policy* of the given goal. As long as the model is powerful enough, the learned USFs can directly generalize to unseen goals. For each such goal, USFs can either produce a $\boldsymbol{\psi}$ corresponding to an optimal policy for that goal, or provide a reasonable approximation.

For the features $\boldsymbol{\phi}(s_t, a_t, s_{t+1})$, as discussed above, it is often sufficient to model it as $\boldsymbol{\phi}(s_{t+1})$. When there exists meaningful, natural representations for states, for example, one-hot vectors in a gridworld, we can use them for state representation. If not then we need to learn $\boldsymbol{\phi}(s)$. One obvious way to learn $\boldsymbol{\phi}(s)$ would be to regress $r$ on $\boldsymbol{\phi}, \mathbf{w}$, but, as we mentioned above, learning based on $r$ is less informative. Unlike Kulkarni et al. (2016) who learn the state representations separately with an auto-encoder, we embed the learning of $\boldsymbol{\phi}$ into the learning of the USFs, in particular, one hidden layer in the neural networks as shown in Fig.1.

---

[3]This method can also be used with any algorithm that learns state-values to learn *successor representations*.

[4]Here we illustrate based on a single transition, but very often we use a mini-batch for training.

[5]Here we focus on optimal control. In case of on-policy evaluation for a given policy $\pi$, the next state action could be $a = \pi(s_{t+1})$ instead.

---

**Algorithm 1** Multi-goal DQN with USFs

---

1: Initialize $\boldsymbol{\theta}$ and replay buffer $\mathcal{D}$
2: **for** each episode **do**
3:      Get goal $g$ and initial state $s_0$
4:      **for** each time step $t = 0, \cdots, T$ **do**
5:          $a_t = \text{Uniform}(|\mathcal{A}|)$ if $\text{Bernoulli}(\epsilon)$ otherwise $\text{argmax}_a Q_g(s_t, a; \boldsymbol{\theta})$       $\triangleright$ $\epsilon$-greedy
6:          Execute $a_t$ and observe next state $s_{t+1}$ and immediate reward $r_{t+1}$
7:          Store transition $\{g, s_t, a_t, s_{t+1}, r_{t+1}, \gamma_{t+1}\}$ in $\mathcal{D}$
8:          Sample random minibatch of transitions $\{g_i, s_i, a_i, s_{i+1}, r_{i+1}, \gamma_{i+1}\}$ from $\mathcal{D}$
9:          Compute target $\widehat{Q}_i$ and $\widehat{\psi}_i$ according to Eq. (3) and (4)
                                       $\triangleright$ as opposed to $y_t$ in Eq. (2) in case of multi-goal DQN
10:       Perform gradient descent on $L$ w.r.t. $\boldsymbol{\theta}$ based on the targets      $\triangleright$ as opposed to Eq. (2)
11:      **end for**
12: **end for**

---

Fig. 1 shows the neural network structure of our USFs. This version corresponds to the finite action case. After going through part of the USF network, a state $s$ is represented by a $d$-dimensional vector $\phi(s)$ (Kulkarni et al., 2016). Then it is concatenated with a goal vector to produce a ($|\mathcal{A}| \times d$)-dimensional SF vector $\psi(s, a, g)$ for all $|\mathcal{A}|$ actions. On the right-most part of the network, we have a model to compute $\mathbf{w}_g$ given a goal $g$. The inner product of $\psi$ and $\mathbf{w}$ will be the action-value $Q$ for the corresponding action. As a concrete example, Algo. 1 shows how DQN can be extended to Multi-goal DQN and how USFs can be applied to Multi-goal DQN.

There is an easy extension for the continuous action case. The base RL algorithm (e.g., DDPG, which also naturally extends to a Multi-goal setting) may have an additional "actor" component for the policy. In such cases, our model, with some minor modifications, can accommodate this. We elaborate on this in Appendix B.

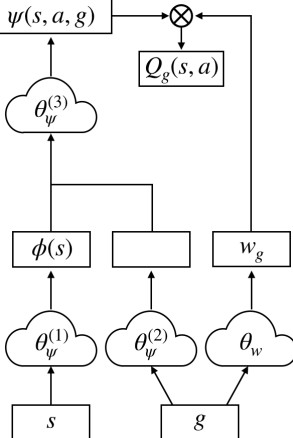

Figure 1: Model Architecture

## 3   EMPIRICAL STUDIES

When evaluating the transfer capability of the USFs framework, we seek to answer two key questions. Firstly, when learning with multiple goals, can USFs successfully transfer knowledge between these goals and consequentially speed up training? Secondly, how well can it transfer knowledge from previous tasks to new, previously unseen tasks? To answer the first question, we begin by training USFs on a set of training tasks (or source tasks) to evaluate its performance under a multi-task setting. Then, to answer the second question, we switch to a new set of unseen tasks (or target tasks) and observe how quickly and how well the agent can adapt to the new set of tasks. In practice, this means we use the parameters the model has learned on the source tasks as initialization for the model on the target tasks. Note that we also carry over the replay buffer from the source tasks.

In order to answer our aforementioned questions, we compare Multi-goal DQN with USFs as described in Algo. 1 against Multi-goal DQN. While this is sufficient for the finite action case, we would also like to consider the continuous action case. For the continuous action case, we note that as with DQN, DDPG (Lillicrap et al., 2016) naturally extends to multiple-goals. We refer to the resulting algorithm as Multi-goal DDPG, and we similarly compare it to Multi-goal DDPG with USFs. Note that as both Multi-goal DQN and Multi-goal DDPG can generalize over goals, they are strong baselines for comparison. We do not compare to UVFA because learning it while exploring the environment in a Reinforcement Learning fashion is prone to be unstable and may require further prior knowledge to achieve reasonable performance (Schaul et al., 2015, Sec. 5.3).

In order to deal with this two-stage procedure mentioned above, we have to treat the replay buffer used in the above algorithms differently during the second stage. In the first stage the replay buffer is treated normally, but during the second stage, we maintain two buffers. The first buffer is the buffer carried over from training on the source tasks. The second buffer is a new buffer where every new

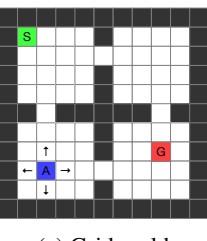
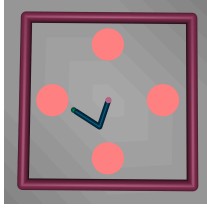
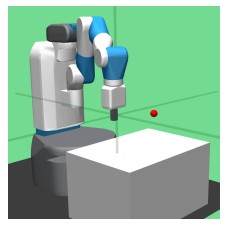
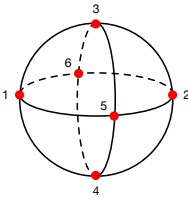

(a) Gridworld      (b) Reacher      (c) Fetch      (d) Fetch Training

Figure 2: (a) Simple gridworld with the start state denoted by $S$, the goal state denoted by $G$ and the agent denoted by $A$. In each state, the agent can move in any of the four cardinal directions, but if it attempts to walk into a wall, it will instead remain in the same state. (b) Simulated limb with two joints that tries to move its tip to a specific goal. The four red circles indicate the training regions. (c) Simulated robotic arm that tries to reach the goal position denoted by a red dot. (4) Training regions for FetchReach-v2.

transition is stored. Whenever we sample from the replay buffer in the second stage, we randomly pick one of the two buffers with equal probabilities.

To evaluate the performance of the above algorithms, we use the *done rate*, or the percentage of episodes where the agent reaches the goal. We also include the average number of steps per episode as a supplementary result in the appendix. We calculate these metrics using the same set of the goals the agent is learning on.

Additionally, in Sec. 3.1, we also evaluate USFs on a set of hold-out goals, which is disjoint from the set of source and target goals. This set of hold-out goals is used to evaluate the direct transfer performance (Taylor & Stone, 2009) or the performance on unseen goals *without learning* from interacting with them. This is in contrast to the target goals where we evaluate the performance on unseen goals when learning from interacting with them.

All our results are averages of 20 runs. We smooth our curves with a mean filter using window size of 50 in our gridworld experiments and a window size of 25 in our MuJoCo experiments. In training we use $\epsilon$-greedy when there are finite actions, but when evaluating we only consider the greedy policy in both the finite action and continuous action settings to measure the actual quality of the learned policy.

## 3.1 GRIDWORLD

We first evaluate USFs on a simple gridworld environment as shown in Fig. 2a, where the agent is trying to reach a specific goal cell. Within this domain we consider three different reward structures:

**constant** $-0.1$ per step, 0 when reaching the goal

**room** $-k/10$ per step, 0 when reaching the goal where $k$ is the number of rooms the agent must still traverse (including the room the goal is in) to reach the goal using the optimal policy

**mixed** at the start of each episode, randomly pick one of the two reward structures above

The room based reward structure is designed to provide the agent with an approximate distance to the goal while reducing the likelihood that the agent will get stuck in corners.

In this problem, as shown in Fig. 1, the network takes the $(x, y)$ coordinates of the current state and goal as input. However, $\phi$, which is the target of the USFs can either be a one-hot indicator vector as discussed in Sec. 2.1 or can be learned as described in Sec. 2.2. We refer to the former as the *one-hot representation* and the latter as the *learned representation*. We limit each episode to 31 steps, and, for each episode, the goal is randomly selected from a set of 12 goals. To reduce the variability of goal sets and subsequently ease analysis, each set of goals is formed by randomly selecting 3 goals in each room without replacement. After 48k timesteps of training in the first stage, the set of 12 goals is swapped out for a new, disjoint set of 12 goals formed by the same method for testing. This allows us to determine the extent to which USFs can perform transfer learning.

We sweep over hyper-parameter settings and report results from the best performing one. Fig. 3 (left) compares the performance when evaluation is done using the same set of goals while Fig. 3 (right)

compares the performance when evaluated on a set of hold-out goals. This helps us analyze the generality of the algorithms or the ability of the algorithms to transfer knowledge to new, unseen tasks *without learning from them*. We provide the corresponding number of steps per episode in Appendix C.

We make several observations regarding the aforementioned results. (1) During training in the first stage, Multi-goal DQN + USFs performs considerably better than Multi-goal DQN in regards to both maximizing the done rate and minimizing the number of steps per episode with all three reward structures. The fact that this improvement is consistent across different reward structures indicates that USFs are robust against different reward structures. (2) For the test stage (second stage denoted by the shaded region), Multi-goal DQN + USFs can adapt at least as fast and occasionally faster. (3) One would expect that because the one-hot representation accurately describes the states and perfectly satisfy Eq. (1), Multi-goal DQN + USFs should perform better using the one-hot representation than with the learned representation. However, this is not always the case. As seen in the constant reward case, using fixed representations can actually decrease performance when compared with learned representations. One reason could be that when the reward structure is more stable/consistent like with the constant reward structure, learning the representations may be a less noisy process and, as a result, more effective. (4) The performance difference with and without USFs is especially notable in Fig. 3 (right) where the greedy policy learned by Multi-goal DQN is evaluated on a set of hold-out goals. This difference in performance means that the USFs can, in fact, generalize better to new goals without even learning them, which may be one reason why it can achieve faster adaptation. (5) When adding USFs to Multi-goal DQN, the improvement is more significant with the room and mixed reward structures than the constant one. This may suggest that USFs are learning the factorization as we described in Sec. 2 and this factorization may be helping the agent deal with reward structures that are very different between goals.

## 3.2 MuJoCo

In order to demonstrate the utility of USFs with continuous state/action spaces, we apply it to two environments that use the MuJoCo physics engine (Todorov et al., 2012). The first, the Reacher-v2 environment (Brockman et al., 2016), is shown in Fig. 2b. The second, OpenAI's Gym FetchReach-v2 environment (Plappert et al., 2018), is shown in Fig. 2c.

For the Reacher-v2 environment, the agent's objective is to move the joints so that the tip of the limb reaches a specific goal location. In this environment, the goal $g$ is specified by its $(x, y)$ coordinates, the state fed to the network as input includes the position and velocity of the joints, and the actions are continuous and provide control of the joints.

In order to build a set of training and test goals for use with the Reacher-v2 environment, we partition the reachable goal space. We use 4 spots around a 2-D platform as shown in Fig. 2b to select training goals. Training proceeds for 45k steps. In the second stage, we randomly select points from the remaining reachable areas for testing.

For the FetchReach-v2 environment, the agent moves a simulated robotic arm and tries to make its gripper reach a specific goal location. The goal $g$ is specified as $(x, y, z)$ coordinates, the state fed to the network as input includes the position and velocity of the hand, and the actions are continuous and provide three-dimensional control of the hand and one-dimensional control of the hand's gripper. To make the task harder, we lower the threshold for when the gripper has reached the goal to $0.04$.

Similarly to the Reacher-v2 environment, to build a set of training and test goals, we partition the reachable goal space. We first place the large sphere shown in Fig. 2d into the reachable area of the arm. We then select the 6 highlighted spots around the sphere to select training goals. Training proceeds for 50k steps. As with the FetchReach-v2 environment, we randomly select points from the remaining reachable areas for testing.

Unlike in the gridworld experiments of Sec. 3.1, in both the Reacher-v2 environment and the FetchReach-v2 environment $\phi$ must be learned. The reward signal for both environments is the negative distance to the goal.

We show the done rate for the Reacher-v2 environment in Fig. 4a and for the FetchReach-v2 environment in Fig. 5a. We furthermore provide the corresponding number of steps per episode in Fig. 9a and Fig. 9b from Appendix C. For both the Reacher-v2 and the FetchReach-v2 environment,

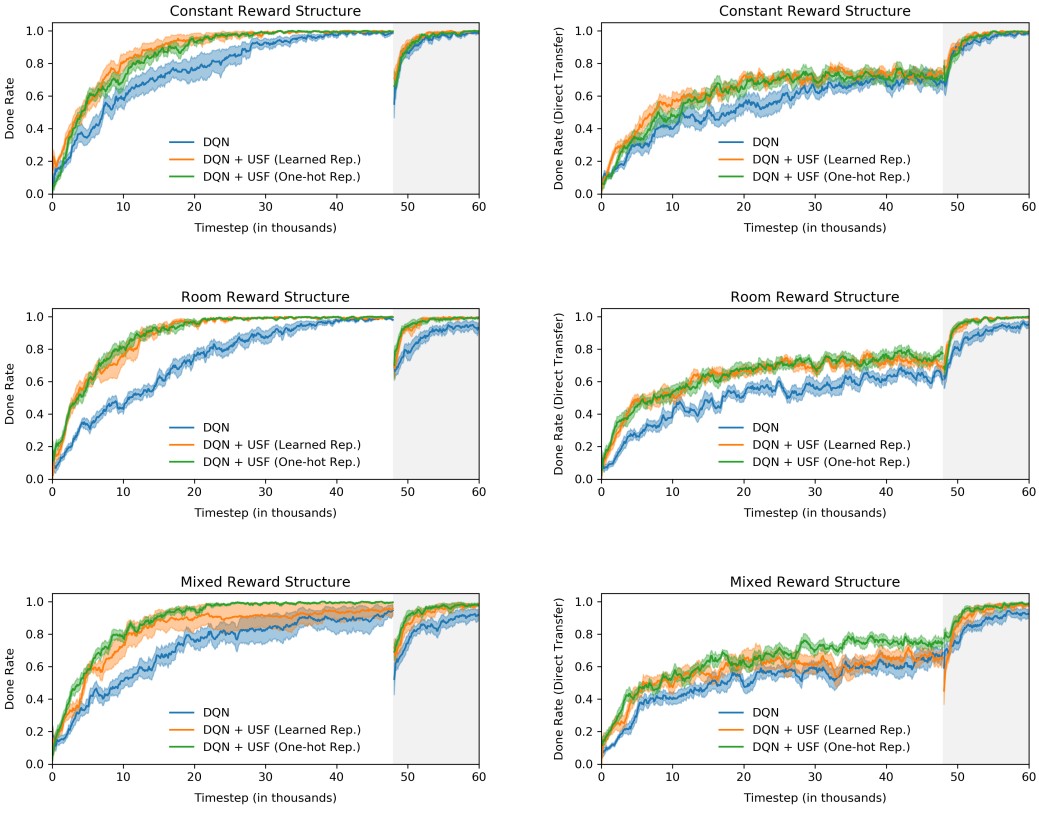

Figure 3: Done rate on the gridworld domain when evaluating on the current set of goals (left) and a tertiary set of goals (right). The three rows correspond to the constant (top), room (middle) and mixed (bottom) reward structures, respectively. Shading denotes standard error.

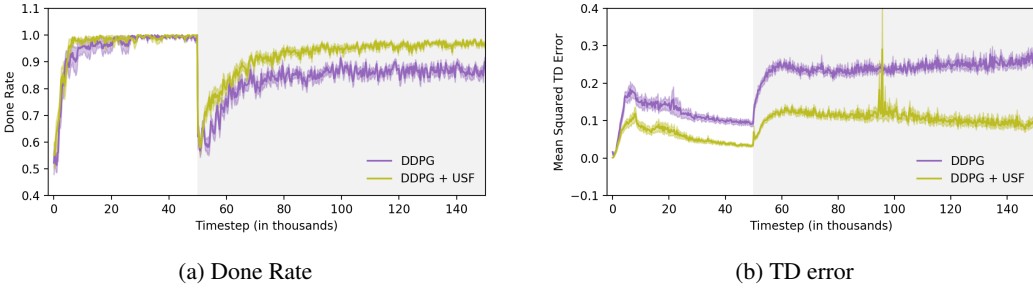

(a) Done Rate

(b) TD error

Figure 4: Done rate & TD error on the Reacher-v2 environment. Shading denotes standard error.

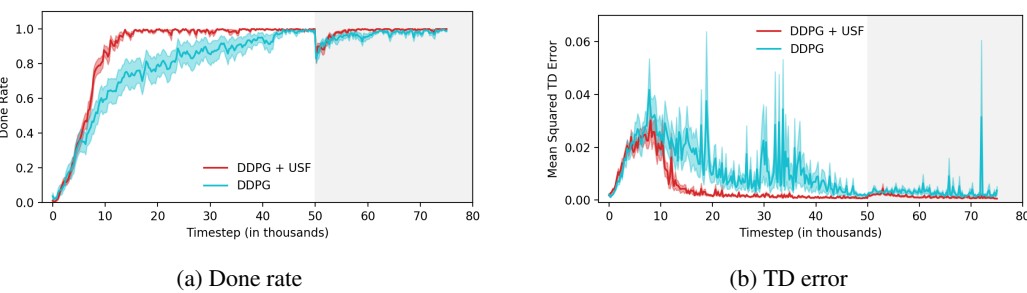

(a) Done rate

(b) TD error

Figure 5: Done rate & TD error on the FetchReach-v2 environment. Shading denotes standard error.

USFs provide a considerable advantage in the first stage, which suggests that USFs are useful when transferring between multiple tasks even in complicated environments. When solving unseen tasks in the second stage, Multi-goal DDPG + USFs again offers an advantage over Multi-goal DDPG. Additionally, in Fig. 4a there is a noticeable gap in the asymptotic performance, which indicates that USFs may be less subject to bias carrying over from previous tasks. One possible reason why Multi-goal DDPG + USFs may perform better is that USFs capture the transition dynamics of the environment, which tend to be stable among tasks. To investigate why there exists such pronounced difference in performance, we also plot their TD errors (Eq. (3)) in Fig. 4b and Fig. 5b. We can observe that Multi-goal DDPG + USFs tends to have a more stable temporal difference error. This may be one of the underlying reasons for more effective learning.

## 4 Related Work

This work is most closely related to Universal Value Function Approximators (Schaul et al., 2015), or UVFA. UVFA uses a two-stage procedure to learn this approximator, primarily in a supervised learning fashion. It begins by constructing a state and goal representation using transitions from multiple tasks/goals via matrix factorization. Then, in the second stage, it approximates such representations using deep neural nets so that it can generalize to unseen goals. One of the key drawbacks of UVFA is that it relies heavily on the availability of known values for existing goals, whether optimal or estimated by a Horde-like architecture (Sutton et al., 2011). Learning UVFA while exploring the environment in a Reinforcement Learning fashion is prone to be unstable and may require further prior knowledge to achieve reasonable performance (Schaul et al., 2015, Sec. 5.3). In contrast, we demonstrate that our method can successfully learn SFs via direct bootstrapping and can be applied on top of any algorithm that learns state-action values using a temporal difference method.

This work is also related to Successor Features (SFs) (Barreto et al., 2017; 2018). Barreto et al. (2017) perform General Policy Improvement (GPI) using previous tasks' values. However, unlike with USFs, tasks are learned in a sequential manner (Caruana, 1997) and there is no direct generalization to unseen tasks. The follow-up work by Barreto et al. (2018) showed that using rewards and action-values from training tasks as state features and successor features respectively is equally expressive when the training tasks are linearly independent. When facing a new task, however, a new set of successor features must be learned, which means the model is required to grow linearly as we encounter more and more goals. In comparison, USFs use a shared successor feature model to handle a large number or even infinitely many goals, thereby avoiding an excessive memory cost.

Using the related concept of Successor Representations (SRs), Kulkarni et al. (2016) proposed a deep learning framework to approximate SRs and incorporate $Q$-learning to learn SRs through interacting with an environment that has a *single* goal. In comparison, our approach learns the universal SFs that generalize not only over states but also over goals. This generalization allows USFs to perform both multi-task learning and transfer learning. Furthermore, while Kulkarni et al.'s method and USFs both make use of TD error, we additionally use $Q$-loss on state-action value (Eq. (3)) instead of reward-prediction loss (from Eq. (1)) to learn both USFs and goal-specific features $\mathbf{w}$. Appendix D show that this modification can lead to a noticeable improvement in performance.

In addition to some of the work mentioned above, there exist several other papers that focus on learning with multiple tasks or goals. Horde (Sutton et al., 2011) learns multiple general value functions simultaneously for preset goals in an off-policy manner. Unlike our method, it cannot handle an infinite number of goals and cannot generalize to unseen new goals. Hindsight Experience Replay (Andrychowicz et al., 2017) maintains a replay buffer with additional goals for each trajectory to speed up learning and improve sample efficiency. It is based on an observation that every failure trajectory (i.e., every episode where the agent was not able to reach its goal) can be a successful one if we treat the final state of the trajectory as the goal. Exploiting this observation creates a method that can generalize over goals when the reward signal is sparse. However, unlike with USFs, HER *requires* access to the reward function for each additional goal. If such information is indeed available, USFs can be used in conjunction with HER. We investigate this in Appendix E. Temporal Difference Models (Pong et al., 2018) learn time-varying value functions for goal-oriented problems. It is somewhat related to our work in that it is also concerned with using temporal difference methods to learn the dynamics of the environment. Finally, UNREAL (Jaderberg et al., 2016) uses

auxiliary tasks to facilitate learning in a multi-goal environment. However, they do not explicitly attempt to generalize over multiple goals or transfer knowledge between tasks.

## 5 CONCLUSION

In this work, we proposed Universal Successor Features (USFs) to perform transfer in Reinforcement Learning when tasks share the same underlying dynamics, but their goals differ. We noted that USFs can be applied on top of any existing algorithms that learn action-values or state-values using a temporal difference method and subsequently presented an end-to-end architecture that can learn USFs by interacting with the environment. Finally, we demonstrated that USFs can accelerate learning and are able to generalize over goals, including unseen ones.

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

## A    MULTI-GOAL BASELINES

In this section, we elaborate on the multi-goal DQN and multi-goal DDPG baselines appearing in Sec. 2 and 3. Algo. 2 shows the pseudo-code for Multi-goal DQN *without* USFs, and the corresponding model architecture is shown in Fig. 6. The two key differences between Algo. 2 and Algo. 1 is that in Algo. 2 the target and the objective function are based on Eq. (2) instead of Eq. (3) and (4). The extension of this to DDPG is straightforward.

---

**Algorithm 2** Multi-goal DQN

---

1: Initialize $\boldsymbol{\theta}$ and replay buffer $\mathcal{D}$
2: **for** each episode **do**
3:     Get goal $g$ and initial state $s_0$
4:     **for** each time step $t = 0, \cdots, T$ **do**
5:         $a_t = \text{Uniform}(|\mathcal{A}|)$ if Bernoulli($\epsilon$) otherwise $\text{argmax}_a Q_g(s_t, a; \boldsymbol{\theta})$          $\triangleright$ $\epsilon$-greedy
6:         Execute $a_t$ and observe next state $s_{t+1}$ and immediate reward $r_{t+1}$
7:         Store transition $\{g, s_t, a_t, s_{t+1}, r_{t+1}, \gamma_{t+1}\}$ in $\mathcal{D}$
8:         Sample random minibatch of transitions $\{g_i, s_i, a_i, s_{i+1}, r_{i+1}, \gamma_{i+1}\}$ from $\mathcal{D}$
9:         Compute target $y_i = r_{i+1} + \gamma_{i+1} \max_a Q(s_{i+1}, a, g_i; \boldsymbol{\theta})$ (see Eq. (2))
10:        Perform (semi-)gradient descent on $[Q(s_i, a_i, g_i; \boldsymbol{\theta}) - y_i]^2$ w.r.t. $\boldsymbol{\theta}$ based on the target
11:    **end for**
12: **end for**

---

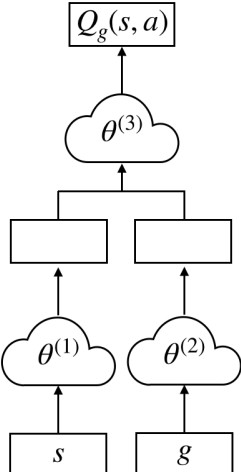

Figure 6: Multi-goal DQN Model Architecture

## B  DDPG WITH USFS

In this section, we discuss how DDPG can be extended to the multi-goal setting and how it can be used with USFs. Algo. 3 shows the pseudo-code for Multi-goal DDPG with USFs, and the corresponding model architecture is shown in Fig. 7. There are two major differences from Algo. 1. First, the action selection method is different. The exploration (line 5) and the next-state action (line 9) now depend on the actor network (dashed components in Fig. 7) instead of the action-values or taking the maximum. Second, there is an additional step of policy gradient (line 12) when updating the actor network.

---

**Algorithm 3** Multi-goal DDPG with USFs

---

1: Initialize USFs' parameters $\boldsymbol{\theta}$, actor parameters $\boldsymbol{\theta}_\pi$ and replay buffer $\mathcal{D}$
2: **for** each episode **do**
3:     Get goal $g$ and initial state $s_0$
4:     **for** each time step $t = 0, \cdots, T$ **do**
5:         $a_t = \pi(s_t; \boldsymbol{\theta}_\pi) + \epsilon$ where $\epsilon \sim \mathcal{N}(0, \sigma^2)$         ▷ Noisy action
6:         Execute $a_t$ and observe next state $s_{t+1}$ and immediate reward $r_{t+1}$
7:         Store transition $\{g, s_t, a_t, s_{t+1}, r_{t+1}, \gamma_{t+1}\}$ in $\mathcal{D}$
8:         Sample random minibatch of transitions $\{g_i, s_i, a_i, s_{i+1}, r_{i+1}, \gamma_{i+1}\}$ from $\mathcal{D}$
9:         Compute next state action $a_{i+1} = \pi(s_{i+1}; \boldsymbol{\theta}_\pi)$
10:        Compute target $\widehat{Q}_i$ and $\widehat{\psi}_i$ according to Eq. (3) and (4) with $a_{i+1}$ instead of $a^*$
11:        Perform gradient descent on $L$ w.r.t. $\boldsymbol{\theta}$ based on the targets
12:        Update $\boldsymbol{\theta}_\pi$ by policy gradient descent
13:     **end for**
14: **end for**

---

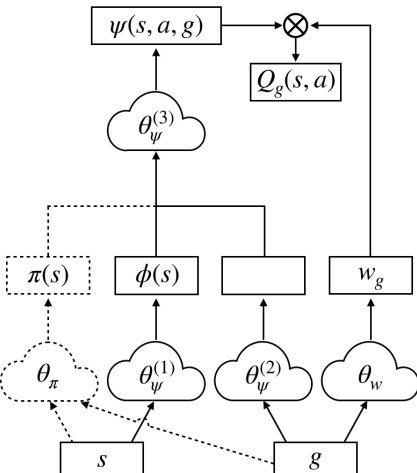

Figure 7: Multi-goal DDPG with USFs Model Architecture

# C  NUMBER OF STEPS PER EPISODE

Fig.8 shows the average number of steps per episode from the runs appearing in Fig.3, Fig.9a shows that for the runs appearing in Fig.4a, and Fig.9b shows that for the runs appearing in Fig.5a.

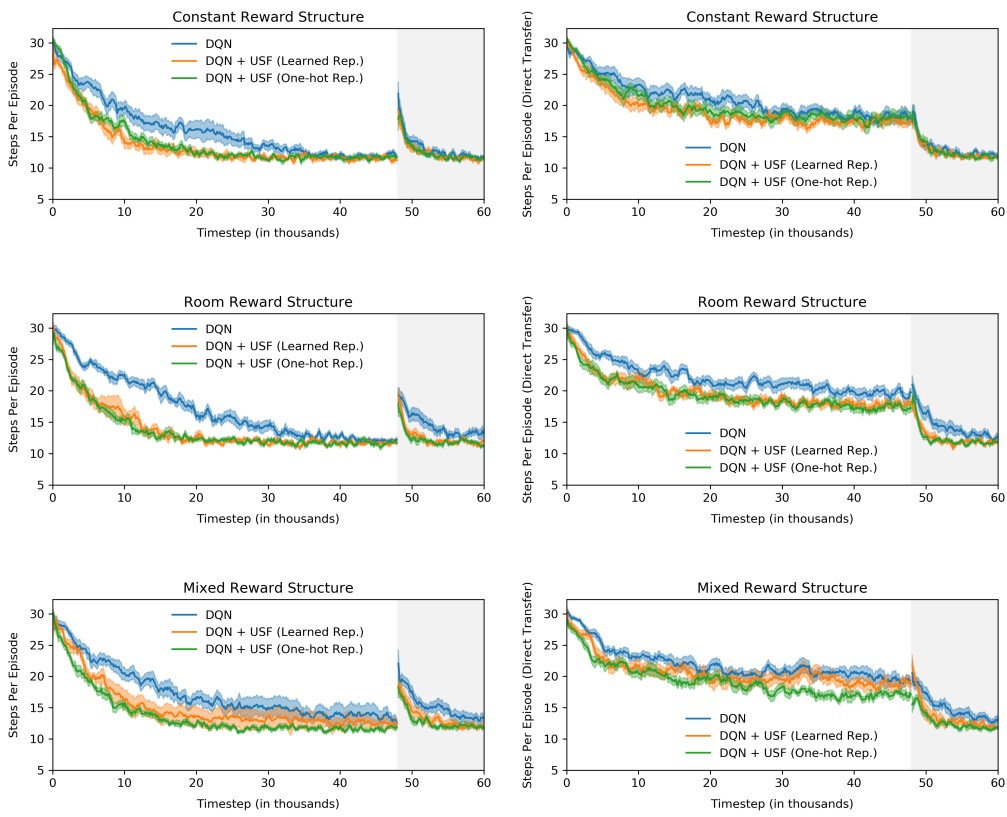

Figure 8: Steps per episode on the gridworld domain evaluating on the current set of goals (left) and a tertiary set of goals (right). The three rows correspond to the constant (top), room (middle) and mixed (bottom) reward structures. Shading denotes standard error.

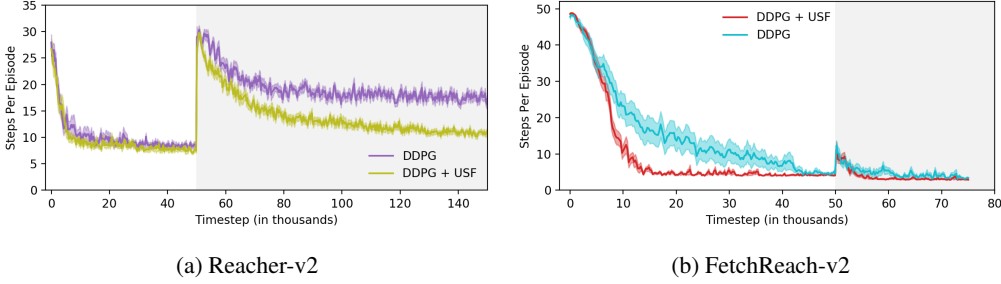

(a) Reacher-v2                              (b) FetchReach-v2

Figure 9: Steps per episode on the two MuJoCo environments. Shading denotes standard error.

# D LEARNING WITH AN ALTERNATIVE OBJECTIVE

This section empirically justifies our proposed objective and showcase the difference in performance when using an obvious alternate objective. Recall that, as described in Sec. 2.2, we use $L_Q + \lambda \cdot L_\psi$ as our objective when learning the model parameters. One could instead use Eq.(1) to replace Eq.(3) with the following loss:

$$L_r = \left[ r_{t+1} - \boldsymbol{\phi}(s_t, a_t, s_{t+1})^\top \mathbf{w}(g; \boldsymbol{\theta}_w) \right]^2 .$$

Fig. 10 shows the done rates on the Gridworld domain described in Sec. 3.1 when using an action-value–based loss ($L_Q + \lambda \cdot L_\psi$) and when using a reward-prediction–based loss ($L_r + \lambda \cdot L_\psi$). Here we observe that, under all three reward structures, $L_Q$ performs considerably better than $L_r$.

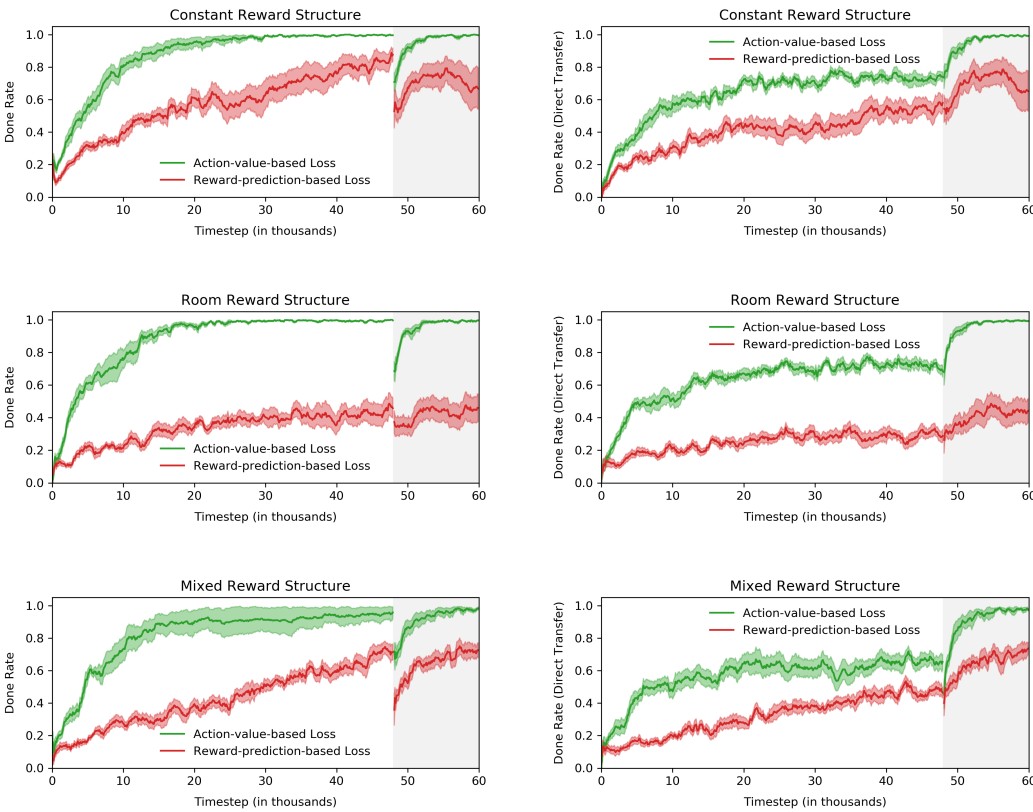

Figure 10: Done rate on the gridworld domain when evaluating on the current set of goals (left) and a tertiary set of goals (right), using two different objective functions. The three rows correspond to the constant (top), room (middle) and mixed (bottom) reward structures, respectively. Shading denotes standard error.

# E  EXPERIMENTS WITH HER

Hindsight Experience Replay (Andrychowicz et al., 2017), or HER, is a method of accelerating learning in multi-goal settings. It accomplishes this by hallucinating experienced transitions as if they were experienced while pursuing an alternate goal. However, notice that unlike USFs, HER requires access to the reward function for the alternative goal in order to hallucinate accurate transitions. If such information is indeed available, USFs can be used in conjunction with HER.

We experiment with USFs and HER in both the gridworld domain described in Sec. 3.1, and the two MuJoCo domains described in Sec. 3.2. Fig. 11 shows the done rate achieved by Multi-goal DQN with and without USFs when they are used in conjunction with HER. Fig. 12 shows the done rate achieved by Multi-goal DDPG with and without USFs when they are combined with HER. In both cases, we observe that HER does not diminish and occasionally increases the benefits of USFs.

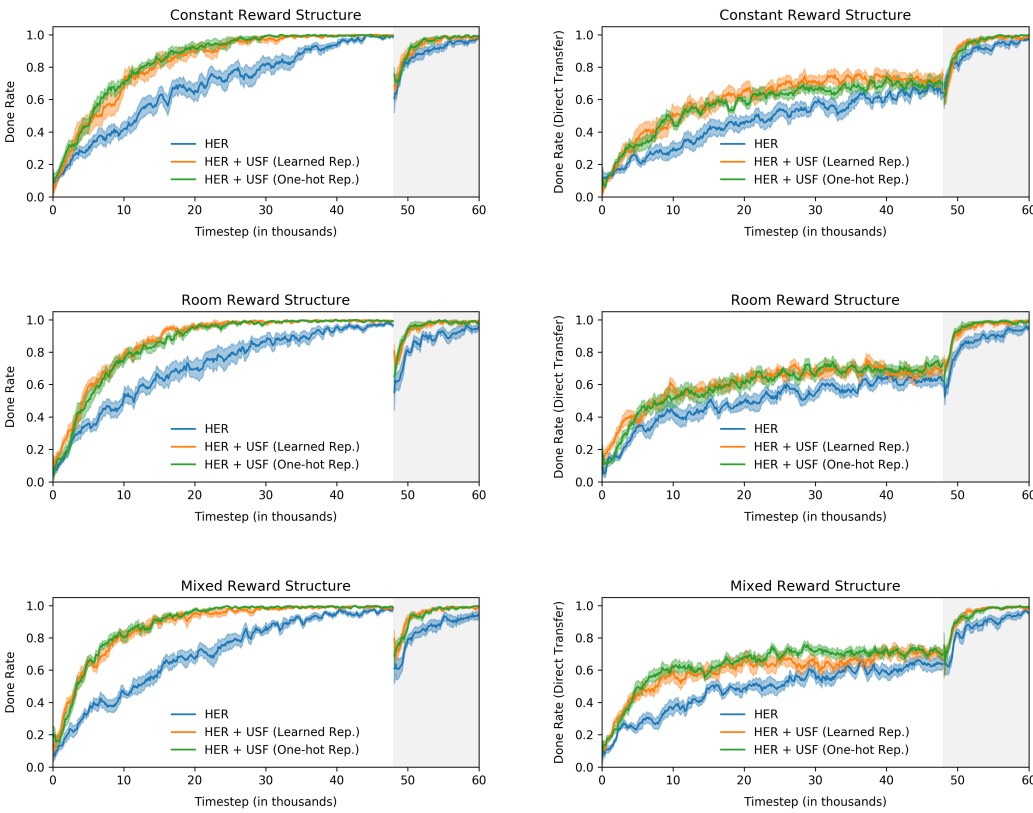

Figure 11: Done rate on the gridworld domain when evaluating on the current set of goals (left) and a tertiary set of goals (right), using HER. The three rows correspond to the constant (top), room (middle) and mixed (bottom) reward structures, respectively. Shading denotes standard error.

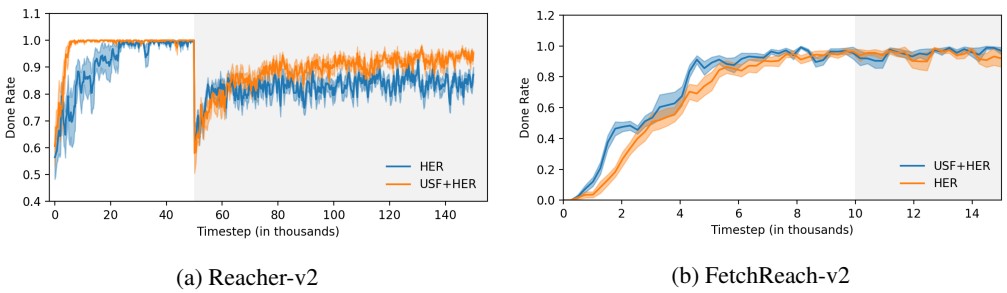

Figure 12: Done rate on the two MuJoCo environments, using HER. Shading denotes standard error.

## F  EXPERIMENTAL DETAILS

In this section, we detail the hyperparameters used and the architectural details of our experiments.

For the Gridworld domain, we used Multi-goal DQN and Multi-goal DQN with USFs. The architecture of Multi-goal DQN appears in 6. Here $\theta^{(1)}$, $\theta^{(2)}$, and $\theta^{(3)}$ were fully connected networks with one hidden layer of 81, 64, and 256 nodes respectively and ReLU activation. The architecture of Multi-goal DQN with USFs is described in App. A and shown in Fig. 6. Here $\theta^{(1)}$, $\theta^{(2)}$, and $\theta^{(3)}$ were fully connected networks with one hidden layer of 81, 64, and 256 nodes respectively and ReLU activation. $\theta_w$ was a fully connected network with two hidden layers of 64 nodes and ReLU activation.

For the MuJoCo domain, we used Multi-goal DDPG and Multi-goal DDPG with USFs. The architecture of Multi-goal DDPG is the same as the architecture appearing in Fig. 6 but with the addition of an actor network. Here the actor network was a fully connected network with two hidden layers of 64 nodes and ReLU activation. $\theta^{(1)}$, $\theta^{(2)}$, and $\theta^{(3)}$ were fully connected networks with one hidden layers of 64 nodes and ReLU activation. The architecture of Multi-goal DDPG with USFs is described in App. B and shown in Fig. 6. Here $\theta_\pi$ and $\theta_w$ were fully connected networks with two hidden layers of 64 nodes and ReLU activation. $\theta_\psi^{(1)}$, $\theta_\psi^{(2)}$, $\theta_\psi^{(3)}$ were fully connected networks with one hidden layer of 64 nodes and ReLU activation.

Table 1: Hyperparameters for Gridworld (Constant)

| Hyperparameter | DQN | DQN + USFs (Learned) | DQN + USFs (One-hot) |
|---|---|---|---|
| Learning Rate | 5e-4 | 5e-4 | 5e-4 |
| Epsilon | 0.25 | 0.25 | 0.25 |
| Loss Weight $\lambda$ | N/A | 0.01 | 0.01 |
| Batch Size | 32 | 32 | 32 |
| Discount Factor $\gamma$ | 0.99 | 0.99 | 0.99 |
| Target Network Update Freq. | 10 | 10 | 10 |

Table 2: Hyperparameters for Gridworld (Room)

| Hyperparameter | DQN | DQN + USFs (Learned) | DQN + USFs (One-hot) |
|---|---|---|---|
| Learning Rate | 5e-4 | 5e-4 | 5e-4 |
| Epsilon | 0.25 | 0.25 | 0.25 |
| Loss Weight $\lambda$ | N/A | 1e-8 | 1e-6 |
| Batch Size | 32 | 32 | 32 |
| Discount Factor $\gamma$ | 0.99 | 0.99 | 0.99 |
| Target Network Update Freq. | 10 | 10 | 10 |

Table 3: Hyperparameters for Gridworld (Mixed)

| Hyperparameter | DQN | DQN + USFs (Learned) | DQN + USFs (One-hot) |
|---|---|---|---|
| Learning Rate | 5e-4 | 5e-4 | 5e-4 |
| Epsilon | 0.25 | 0.25 | 0.25 |
| Loss Weight $\lambda$ | N/A | 1e-6 | 1e-6 |
| Batch Size | 32 | 32 | 32 |
| Discount Factor $\gamma$ | 0.99 | 0.99 | 0.99 |
| Target Network Update Freq. | 10 | 10 | 10 |

---

[1]the probability of sampling from the buffer of hallucinated transitions

Table 4: Hyperparameters for Gridworld (Constant), using HER

| Hyperparameter | HER | HER + USFs (Learned) | HER + USFs (One-hot) |
|---|---|---|---|
| Learning Rate | 5e-4 | 5e-4 | 5e-4 |
| Epsilon | 0.25 | 0.25 | 0.25 |
| Loss Weight $\lambda$ | N/A | 1e-8 | 1e-6 |
| Batch Size | 32 | 32 | 32 |
| Discount Factor $\gamma$ | 0.99 | 0.99 | 0.99 |
| Target Network Update Freq. | 10 | 10 | 10 |
| HER Future Steps | 30 | 30 | 30 |
| HER Buffer Sampling Probability[1] | 0.5 | 0.5 | 0.5 |

Table 5: Hyperparameters for Gridworld (Room), using HER

| Hyperparameter | HER | HER + USFs (Learned) | HER + USFs (One-hot) |
|---|---|---|---|
| Learning Rate | 5e-4 | 5e-4 | 5e-4 |
| Epsilon | 0.25 | 0.25 | 0.25 |
| Loss Weight $\lambda$ | N/A | 1e-8 | 1e-6 |
| Batch Size | 32 | 32 | 32 |
| Discount Factor $\gamma$ | 0.99 | 0.99 | 0.99 |
| Target Network Update Freq. | 10 | 10 | 10 |
| HER Future Steps | 30 | 30 | 30 |
| HER Buffer Sampling Probability[1] | 0.5 | 0.5 | 0.5 |

Table 6: Hyperparameters for Gridworld (Mixed), using HER

| Hyperparameter | HER | HER + USFs (Learned) | HER + USFs (One-hot) |
|---|---|---|---|
| Learning Rate | 5e-4 | 5e-4 | 5e-4 |
| Epsilon | 0.25 | 0.25 | 0.25 |
| Loss Weight $\lambda$ | N/A | 1e-4 | 0.01 |
| Batch Size | 32 | 32 | 32 |
| Discount Factor $\gamma$ | 0.99 | 0.99 | 0.99 |
| Target Network Update Freq. | 10 | 10 | 10 |
| HER Future Steps | 30 | 30 | 30 |
| HER Buffer Sampling Probability[1] | 0.5 | 0.5 | 0.5 |

Table 7: Hyperparameters for Gridworld (Constant), with Alternate Objective

| Hyperparameter | Action-value–based Loss | Reward-prediction–based Loss |
|---|---|---|
| Learning Rate | 5e-4 | 5e-4 |
| Epsilon | 0.25 | 0.25 |
| Loss Weight $\lambda$ | 0.01 | 0.01 |
| Batch Size | 32 | 32 |
| Discount Factor $\gamma$ | 0.99 | 0.99 |
| Target Network Update Freq. | 10 | 10 |

Table 8: Hyperparameters for Gridworld (Room), with Alternate Objective

| Hyperparameter | Action-value–based Loss | Reward-prediction–based Loss |
|---|---|---|
| Learning Rate | 5e-4 | 5e-4 |
| Epsilon | 0.25 | 0.25 |
| Loss Weight $\lambda$ | 1e-4 | 1e-8 |
| Batch Size | 32 | 32 |
| Discount Factor $\gamma$ | 0.99 | 0.99 |
| Target Network Update Freq. | 10 | 10 |

Table 9: Hyperparameters for Gridworld (Mixed), with Alternate Objective

| Hyperparameter | Action-value–based Loss | Reward-prediction–based Loss |
|---|---|---|
| Learning Rate | 5e-4 | 5e-4 |
| Epsilon | 0.25 | 0.25 |
| Loss Weight $\lambda$ | 1e-4 | 1e-6 |
| Batch Size | 32 | 32 |
| Discount Factor $\gamma$ | 0.99 | 0.99 |
| Target Network Update Freq. | 10 | 10 |

Table 10: Hyperparameters for Reacher-v2

| Hyperparameter | DDPG | DDPG + USFs | HER | HER + USFs |
|---|---|---|---|---|
| Actor Learning Rate | 1e-4 | 1e-4 | 1e-3 | 1e-4 |
| Critic Learning Rate | 1e-3 | 1e-3 | 1e-4 | 1e-3 |
| Loss Weight $\lambda$ | N/A | 1e-4 | N/A | 0.01 |
| Batch Size | 64 | 64 | 64 | 64 |
| Discount Factor $\gamma$ | 0.99 | 0.99 | 0.99 | 0.99 |
| HER Future Steps | N/A | N/A | 50 | 50 |
| HER Buffer Sampling Probability | N/A | N/A | 0.5 | 0.5 |

Table 11: Hyperparameters for FetchReach-v2

| Hyperparameter | DDPG | DDPG + USFs | HER | HER + USFs |
|---|---|---|---|---|
| Actor Learning Rate | 1e-4 | 1e-4 | 1e-4 | 1e-4 |
| Critic Learning Rate | 1e-3 | 1e-3 | 1e-4 | 1e-3 |
| Loss Weight $\lambda$ | N/A | 1e-3 | N/A | 1e-4 |
| Batch Size | 64 | 64 | 64 | 64 |
| Discount Factor $\gamma$ | 0.99 | 0.99 | 0.99 | 0.99 |
| HER Future Steps | N/A | N/A | 50 | 50 |
| HER Buffer Sampling Probability | N/A | N/A | 0.5 | 0.5 |

