# OpenReview forum: "Universal Successor Features for Transfer Reinforcement Learning"
_ICLR.cc/2019/Conference_

### Official Review · AnonReviewer3 · 2018-11-03
**Interesting and clear, but contribution small and with many experimental omissions.**

**Rating:** 6
**Confidence:** 5

**Review:**

In this paper the authors propose an extension to successor features (SF). Akin to UVFAs, they condition on some goal state by concatenating to the current state after some shared preprocessing. The authors claim three contributions: 1) introducing the USF, 2) proposing an appropriate deep learning architecture for it, and 3) showing experimentally that USFs improve transfer both within a goal set and to novel goals.

Claims 1) and 2) don't seem particularly noteworthy. Extending SF to be goal-conditioned is very straightforward, doesn't leverage anything unique to the SF formalism (e.g. the reward weights w already encode a goal in some sense), and doesn't attempt to extend its theoretical grounding. The architecture is likewise unsurprising, and the lack of ablations or alternatives make it seem rather unmotivated.

The usage of a Q-learning loss instead of a reward-prediction loss for updating phi is mentioned without citation. This seems quite novel, and could be a significant contribution if its advantage was demonstrated experimentally.

The experiments appear to show a significant advantage for USFs. For the training-goal-set advantage, it would be useful to know the architecture of multi-goal DQN. One hypothesis is that the extra weight-sharing is what is giving USFs an edge, and this should be ruled out. It is briefly mentioned that UVFAs weren't considered due to their stated instability, but its unclear how they differ from the multi-goal DQN.

The novel-goal results are impressive at first glance, but there is a glaring omission. Hindsight experience replay (HER) is mentioned but not evaluated, and would very likely trivialise the train/test goal-set distinction (unless the test goals were never previously visited). As these results are the primary contribution of this paper, this must be addressed prior to publication acceptance.

Edit: The addition of HER experiments push this up a bit (5-->6). I'm still concerned about how significant the contribution is (as it is a straightforward extension to SFs), but the empirical results are now quite strong.

---

> ### Author Response · Authors · 2018-11-11
> **Request for Minor Clarification**
>
> Thanks for your feedback. We have two brief questions regarding your comments before we can feel confident addressing all your comments later in our full rebuttal. Firstly, when you stated we were using a Q-learning loss instead of reward-prediction loss do you mean we're using Eqn. (3) and (4) as our objective rather than Eqn. (1) and (4)? Secondly, can you elaborate what you mean by extra weight sharing? We've added a short section to the appendix elaborating on our Multi-goal DQN baseline which we hope will help you clarify your comment.

---

> > ### Comment · AnonReviewer3 · 2018-11-11
> > **Clarification**
> >
> > Indeed, I was referring to using Eqn. (3) and (4) instead of (1) and (4). Specifically, Eqn. (1) is normally the only thing optimizing phi (barring an aux loss like reconstruction). Do both (3) and (4) optimize phi, or are there some stop gradients?
> >
> > The weight sharing / weight tying refers to Figure 1, whereby theta_psi is used to embed both the goal and the state. It also looks like theta_psi is used to combine phi(s) and phi(g), but I imagine that is a typo or I'm misinterpreting the figure. The root of my question is basically wondering if the architecture, rather than the losses associated with SF, are what is improving performance. Clarifying the architecture used in each condition would help -- perhaps they are all similar enough that ablations wouldn't be needed.

---

> > > ### Author Response · Authors · 2018-12-04
> > > **Reply to minor clarification**
> > >
> > >
> > > Thank you for your feedback. We also included this answer in our main rebuttal:
> > >
> > > Both (3) and (4) optimize phi. There is no weight sharing in the methods described in the paper. We have changed the notations in Figure 1, 6 and 7 to reflect this. Sorry for the confusion. Appendix A now provides both a detailed description and the model architecture of the Multi-goal DQN baseline. Essentially the USFs architecture is built upon the Multi-goal DQN architecture by replacing the action value with successor features and adding a component to learn w.
> > >
> > > We have attempted to address your concerns in the main rebuttal posted under your original review. Please let us know if there are any other questions or concerns you would like us to respond to.

---

> ### Author Response · Authors · 2018-11-27
> **Thank you for your insightful review and important observations!**
>
> [Novelty about extending SFs to be goal-conditioned; model architecture]
> Appendix A now provides both a detailed description and the model architecture of the Multi-goal DQN baseline. Essentially the USFs architecture is built upon the Multi-goal DQN architecture by replacing the action value with successor features and adding a component to learn w.
>
> There is no weight sharing in the methods described in the paper. We have changed the notations in Figure 1, 6 and 7 to reflect this. Sorry for the confusion.
>
> [Q-learning loss and reward-prediction loss as training objectives]
> We have now added Appendix D which provides an empirical comparison of learning using Eq.(1)+(4) versus learning using Eq.(3)+(4) with the same USFs architecture. We hope this comparison can justify our proposed objective and show the importance here of using a loss based on action values instead of reward-prediction.
>
> [Comparison to UVFA]
> As we discussed with Reviewer#2, the original UVFA uses a two-stage learning procedure, which is unstable in practice. However, there seems to be a common adoption of UVFA which uses end-to-end learning and a goal as input. This is our Multi-goal DQN baseline. We believe that this is a fair proxy for UVFA and by comparing against it, we show the advantage of our method over UVFA.
>
> [Results will be less impressive when using HER?]
> One issue with HER is that it requires access to or must estimate multiple reward functions in each transition. Such a setting is fundamentally different from the setting we evaluate USFs in. However, we note that USFs can be used in conjunction with HER. As we now show in Appendix E, USFs can provide a considerable improvement over HER alone.

---

> > ### Comment · AnonReviewer3 · 2018-12-11
> > **HER applicability**
> >
> > I agree that HER requires being able to evaluate arbitrary-goal-completion rather then just having the environment evaluate the given-goal. However, I don't believe I've ever come across this distinction in the literature, as it seems rare to have access to both the goal representation and given-goal evaluation but not arbitrary-goal evaluation.
> >
> > All of your clarifications were quite informative, but I'm afraid I'm still not convinced by the overall research contribution.

---

> > > ### Author Response · Authors · 2018-12-15
> > > **Clarification to HER applicability**
> > >
> > > While we acknowledge that the HER setting (goal representations with arbitrary-goal evaluation) is common in the literature, our work was primarily influenced by the original, also common UVFA setting (goal representations with given-goal evaluation). In addition to the UVFA paper, there exist several other works that consider the UVFA setting (see for examples [1] and [2]). However, to ensure a fair evaluation of our method, we did compare it both under the UVFA setting and, following your original observation, under the HER setting (see Appendix E where we compare HER with HER + USFs). Our results indicated that USFs may improve performance in this setting as well.
> > >
> > > [1] Zhu, Yuke, et al. "Target-driven visual navigation in indoor scenes using deep reinforcement learning." Robotics and Automation (ICRA), 2017 IEEE International Conference on. IEEE, 2017.
> > > [2] Mankowitz, Daniel J., et al. "Unicorn: Continual Learning with a Universal, Off-policy Agent." arXiv preprint arXiv:1802.08294 (2018).

---

> > > > ### Comment · AnonReviewer3 · 2018-12-18
> > > > **HER experiment details**
> > > >
> > > > Thank you for the additional experiments. I must say I'm quite surprised by HER's poor performance. There is a large performance gap between the current and tertiary goals despite HER's ability to evaluate arbitrary goal-performance off-policy. Indeed, it looks as though this gap is as high for HER as it is for the methods that don't exploit knowledge of the goal-reward function. Could you explain why this is the case? Does the gap shrink when training under a more exploratory policy (e.g. high epsilon)?

---

> > > > > ### Author Response · Authors · 2018-12-20
> > > > > **HER experiment explanation**
> > > > >
> > > > > Indeed, here HER does not appear to offer a significant improvement when compared with Multi-goal DQN. There are several possible reasons. (1) In our experiments, the tertiary goals are not necessarily on the optimal path of any training goal (i.e., the tertiary goals have a chance of never being experienced during training). (2) There is a chance that the system will train on an irrelevant goal (that is not in any of the goal sets), which is not true in the non-HER case. This can potentially outweigh the benefits of using HER in terms of the performance on the tertiary goals. (3) Even when the tertiary goal appears in some training episodes, such a goal may not be sampled from the buffer very frequently. In our experiment, half of the data we train on is from the current goal, while the other half is from goals used with HER. The probability of one particular tertiary goal being sampled is not very high. (4) The results appearing in the paper use an epsilon of 0.25. Further increasing this can potentially degrade the training performance and significantly increase training time, which most likely outweighs the benefits delivered by HER. Nevertheless, our experiments on all three environments suggest that, in the provided environments, USF can work well whereas HER seems to struggle considerably more.

---

### Official Review · AnonReviewer1 · 2018-11-06
**Interesting but not enough depth**

**Rating:** 7
**Confidence:** 5

**Review:**

I like the idea of Universal Successor Features, it seems a bit incremental but I think it is worth exploring. There is some missing aspects and better comparison that can be made for the paper. I believe for the final camera ready these comparisons should be added. Specifically the related work section seems to not be in a desired depth.

From experiments perspective, there is sufficient experiments that can demonstrate the value of the model. It is a simple model but an elegant application and correctly used for the purpose of the tasks in the paper. I have the following questions which their answers may be good additions to the paper:

1. Have you tried analyzing what successor features and goal-specific features learn? For example, one point of addressing this is: what does the agent seem to avoid or do, under your framework (but not normal DQN).
2. The tasks in this paper seemed a bit simplistic, how does the model work on more complex applications (games)? It is hard to establish proper comparison, even though your claims are sufficiently supported.
3. What is your explanation of cases where blue is under green? One could assume they would meet eventually like top-left in Figure 3.

I strongly suggest a rewrite of the related works section and a redo of the graphics. Using PDF may help with odd aspect ratio for text (Fig 4).

---

> ### Author Response · Authors · 2018-11-27
> **Thank you for your insightful review and thought-provoking questions!**
>
> 1. 2. We did analyze what Multi-goal DQN and USFs learn in our domains, but we concluded that our environments are ill-suited to provide a particularly interesting answer to this question; we have generally observed USFs learning faster than DQN, but in these domains, we have not found any strong characteristic differences between the eventual policies they develop. We hope to later apply this method to more complicated domains such as the Arcade Learning Environment, but, while we are excited about the prospects of how USFs might behave when applied to such domains, we felt that our limited computational resources would be better used to provide a fairer comparison with our baseline. As such we opted to leave the more complicated domains as future work.
>
> 3. As to why we believe that Multi-goal DQN with USFs outperforms vanilla Multi-goal DQN, first note that in the first phase of each of our experiments the setting amounts to multi-task learning. So knowledge is being transferred in both the first and second phase of our experiments. In general, we believe that when decomposing the action-values into successor features and goal-specific features, the dynamics of the world learned by the successor features transfer more easily between goals than the action-values alone. We hypothesize that the more dissimilar the values of proximal states are under different goals, the more this dissociation benefits the transfer process. This hypothesis is supported by the larger gap in performance under the room reward structure than under the constant reward structure.
>
> Following your suggestion, we have rewritten the related work section in an attempt to give a more comprehensive and a more careful analysis of prior work. Also, thank you for bringing the graphical issue to our attention. We have redrawn the figures accordingly.

---

### Official Review · AnonReviewer2 · 2018-11-07

**Rating:** 4
**Confidence:** 5

**Review:**


Summary: This paper proposes a generalisation of the SFs framework to a goal conditioning representation that could, in principle, generalise over a collection of goals at test time. This is akin to universal value functions [1] (and more generally GVFs). Although I like the idea  and it seems a very interesting direction for generalisation to new goals, I do think the execution, the particular instantiation and (lack of) in-depth evaluation with (at least some of the) existing methods in literature -- including UVFAs [1] and the different ways SFs have been used for generalisation [2,3,4] -- is unfortunately letting it down.


Clarity: Reasonably well-written, easy to follow. A couple of things in the experimental section can be improved:
- It’s not totally clear to me what the their baseline Multi-goal DQN is. Does it have the same architecture as Figure 1, but just using (2).
- In the plots, the only difference between DQN and DQN+USF is that the second as the additional loss L_{\psi} ? Or is there any other difference?


Originality and Significance:
I’m a bit split here: I like in principle the idea, but I think this instantiation is (fairly) incremental with respect to the current literature. Even the claimed contributions are a bit thin. Suitability of SFs to any TD-based learning, comes from SRs/SFs satisfying a Bellman eq. which was point out, explored and paired with control algorithms before [2,4]. Also, the particular way of learning the features \phi, without going through the rewards, was already proposed and explored in [3]. That might be a missing reference.

The experiments seems to show slight improvements with respect to a baseline (Multi-DQN). It is not clear to me exactly what this is or if it would dominated even something vanilla UVFAs. I think this is a missing and somewhat mandatory comparison. I know the authors noted that is was because ‘UVFAs are prone to instabilities and may require further prior knowledge’, but I think that refers only to the two-stage (factorisation) procedure proposed in the original paper, not the common adoption in the literature. At the end of the day, the proposed architecture in Fig. 1 is a kind of UVFA, just with a bit more structure, so it would be surprising to me if UVFAs would actually fail in these environments. But if that’s the case, that’s a very interesting data point that the additional structure actually helps considerably beyond the incremental advantage exemplified here.


Other comments/questions:

1) Clarification on the training procedure. The value function $Q(s,a,g)$  are training via eq. (3) with the i) actual reward (coming from the environment) or ii) the ‘fictitious’ reward coming from r(s,a,s’|g) = \phi(s,a,s’)^T w(g)? Note that these are very different and only one ensures compatibility between the rewards and the value functions in learning.
The SFs will give you the value function for the reward r(s,a,s’|g) = \phi(s,a,s’)^T w(g) and if this is not align with the real reward, the corresponding value function obtained via SFs will not be the value function optimising the real reward. As far as I can see there’s not criteria that forces this to be the case.

2) Comparison with SF transfer literature. Although discussed in the related work section, there is no quantitative comparison to the way SFs were shown to transfer knowledge[2,4], via evaluation and (generalised) policy improvement. Because these ways of generalisation are very different, it’s not clear go they would stack against each other, or in which scenarios one would be more appropriate than the other.
To give a more concrete example: The training procedure in 3.1 makes sure that there’s fairly good coverage of the whole state-space by sampling goals conditioned on the room. Now if one would train SFs on these train tasks only (even independently), we would have policies that would know how to go to any of the rooms. And for the test tasks we would have the evaluation of these policies to the collection of goals. Which means that applying the methodology of transfer in [2,4] we would zero-shot get policies that reach any of the states encountered in the path of the 12 goals used in the train phase. And even if the test goals are not part of this collection, it stands to reason that a policy that can already go to the goal’s room and be easily adaptable to reaching the test goal -- aka the evaluation the policy that already reached that room is a good starting point for the improvement step [4].

Note: I am willing to reconsider when/if the above have been reconciled/resolved.

References:
[1] Schaul, T., Horgan, D., Gregor, K. and Silver, D., 2015, June. Universal value function approximators. In International Conference on Machine Learning (pp. 1312-1320).

[2] Andre Barreto, Will Dabney, Remi Munos, Jonathan J Hunt, Tom Schaul, Hado P van Hasselt, and ´ David Silver. Successor features for transfer in reinforcement learning. In Advances in Neural Information Processing Systems, pp. 4055–4065, 2017.

[3] Machado, M.C., Rosenbaum, C., Guo, X., Liu, M., Tesauro, G. and Campbell, M., 2018. Eigenoption Discovery through the Deep Successor Representation, International Conference on Learning Representations, 2018.

[4] Barreto, A., Borsa, D., Quan, J., Schaul, T., Silver, D., Hessel, M., Mankowitz, D., Zidek, A. and Munos, R., 2018, July. Transfer in deep reinforcement learning using successor features and generalised policy improvement. In International Conference on Machine Learning (pp. 510-519).

---

> ### Author Response · Authors · 2018-11-11
> **Request for Minor Clarification**
>
> Thanks for your feedback. We have one quick question regarding one of your comments before we can feel confident addressing all your comments later in our full rebuttal. From what we've observed in the literature we've been lead to believe that the common adoption in the literature for UVFA amounts to the Multi-goal DQN baseline we use in our work. As you pointed out, we failed to adequately explain what the algorithm we're calling Multi-goal DQN is. As such we've added a short section to the appendix elaborating on our Multi-goal DQN baseline. Can you confirm that this is the common adoption of UVFA that you would have expected we would compare with? If not is there any chance you could direct us to some work that describes the common adoption?

---

> > ### Comment · AnonReviewer2 · 2018-11-20
> > **Clarification**
> >
> > Thanks for the addition & clarification -- it is indeed helpful! And yes, what I meant by 'common adoption' is end-to-end training of an architecture like the one in Figure 6 (appendix).
> >
> > p.s. Just a minor follow up clarification on the training protocol for Alg. 2: was the training done on-goal (line 7 & seems to suggest that, but just checking)?

---

> > > ### Author Response · Authors · 2018-12-04
> > > **Reply to minor clarification**
> > >
> > >
> > > Thank you for your feedback. We also included this answer in our main rebuttal:
> > >
> > > If you mean that the goal appears in the transition then yes. However, the training mini batch is sampled directly from the replay buffer in which case the transitions sampled are likely not to share a goal.
> > >
> > > We have attempted to address your concerns in the main rebuttal posted under your original review. Please let us know if there are any other questions or concerns you would like us to respond to.

---

> ### Author Response · Authors · 2018-11-27
> **Thank you for your thorough review and insightful questions!**
>
> Clarity:
>
> [Unclear description about the Multi-goal DQN baseline and difference of our method]
> In response to your comment, we have added a detailed algorithm description and the model architecture of the Multi-goal DQN baseline to Appendix A. The most notable difference between Multi-goal DQN and our method is that Multi-goal DQN directly learns the action values, while USFs learn successor features and goal-specific features whose inner product estimates the action values. This difference results in both a markedly different architecture in addition to a distinct loss function.
>
>
> Originality and Significance:
>
> [Successor Features have appeared before]
> Both [2,4] do indeed use SFs for control tasks but, unlike ours, their method does not perform direct generalization. The GPI theorem only guarantees that the current policy is as good as all previous policies and that the agent will improve upon it, instead of directly generalizing from previous policies. For more detailed comparison, please refer to the comment #2 for "other comments/questions".
>
> [This way of learning \phi has been proposed before]
> To clarify, the specific way of learning \phi using one layer of neural networks is, to the best of our knowledge, originally proposed in DeepSR (Kulkarni et al. 2016), which we have explicitly cited in our revised paper (Sec.2.2). Notably, differing from both [3] and DeepSR, we do not use an auto-encoder.
>
> [Comparison with UVFA]
> As we noted previously, and also according to Mankowitz et al. [5], the Multi-goal DQN baseline in our paper can be considered as a common adoption of UVFA. Therefore, we have already shown the advantage of USFs over UVFA to some degree.
>
>
> Other comments/questions:
>
> 1. We use the actual reward, not a fictitious reward generated by our model.
> 2. To compare with the SFs & GPI appearing in [2,4], we have attempted to apply the algorithm in [4] to our gridworld setting, but unfortunately, it fails to learn during the test phase (second stage). Recall that in our experiment, the training phase amounts to a multi-task setting, and test phase is also a multi-task setting. So transfer must occur from a set of tasks to another set of tasks. SFs & GPI is specifically restricted to transferring between a multi-task setting to a single test task. As such it will naturally fail in our test phase.
>
> To elaborate on this a bit further, a critical difference between our USFs and the framework of SFs & GPI in [2,4] is that SFs & GPI depends on the assumption that SFs are only a function of the policy and not of the task/goal (see Eq.(4) in [4]). If this assumption does not hold then applying SFs learned from one goal directly to another goal (as in GPI) can be problematic. For example, consider our gridworld environment in which an episode ends after the agent reaches the goal. Using one-hot state features and under the optimal policy, the true SFs \psi^{\pi_1} will include nothing except for an optimal path from the current state to the goal state g_1 (i.e., only the cells on the path will be non-zero). When we use these SFs for a different goal g_2, \psi^{\pi_1} is no longer accurate because the agent has to continue moving after reaching g_1. In other words, \psi^{\pi_1} fails to represent the true future state visitations when we deploy \pi_1 for g_2. If we update all the SFs as done in [2], then these SFs will be specific to one particular test goal. As a result, SFs & GPI can only deal with one test task at a time and so is not suitable for our experimental setting, in which we will encounter a different test goal in *each episode*. In contrast to this, USFs depend on the goal g through the discount function \gamma_g (see last equation on page 2). Thus, USFs can automatically be adjusted according to a goal, even under the same policy.
>
> One additional difference between USFs and the SFs & GPI framework from [2,4], is that, unlike USFs, the SFs & GPI cannot actually produce a good zero-shot policy. In order to have a zero-shot policy, we have to know the goal-specific features w. In our model, we have a component specifically for computing w given g (see the right-most part of Fig.1). However, in [2,4], given a new goal, w is first randomly initialized and then gradually improved while exploring the new task. While we can certainly use this random w and GPI to compute a zero-shot policy, the resulting policy would be no better than random.
>
>
> Follow up clarification:
>
> If you mean that the goal appears in the transition then yes. However, the training minibatch is sampled directly from the replay buffer in which case the transitions sampled are likely not to share a goal.
>
>
> [5] Mankowitz, Daniel J., et al., Unicorn: Continual Learning with a Universal, Off-policy Agent. arXiv preprint arXiv:1802.08294 (2018).

---

### Author Response · Authors · 2018-11-27
**We thank the reviewers for their valuable feedback and constructive suggestions.**

With the advice of the reviewers, we added the following contents to the original submission:
- We have provided an illustration of the architecture and pseudo-code the Multi-goal DQN baseline (Appendix A).
- We have empirically justified our objective function by comparing it with an obvious alternative (Appendix D).
- We have provided some additional experiments demonstrating the compatibility of USFs with Hindsight Experience Replay (Appendix E).
- We have provided more experimental details and the hyperparameters we used for these experiments (Appendix F).
- We have rewritten the related work section in an attempt to be more comprehensive and to provide a more careful analysis of prior work.

We would like to emphasize that USFs learn the dynamics of the environment under *optimal* policies for the given goals. While the original SFs for each goal can capture certain knowledge about the underlying dynamics of the specific task, USFs are able to learn the shared dynamics.

---

### Public Comment · ~Shane_Gayal1 · 2018-12-15
**Can this method work with Actor - Critic Methods ?**

Can we calculate the Advantage function with this architecture and use it with A3C?

---

> ### Author Response · Authors · 2018-12-16
> **Yes, USFs can work with general actor-critic methods.**
>
> ```Thanks for your interests. Yes, USFs can work with general actor-critic methods. Please refer to the MuJoCo experiments in Sec.3.2 with DDPG where an "actor" component is included for the policy. USFs can be similarly applied to the computation of the Advantage in A3C. The modification is straightforward with our method.```

---

> > ### Public Comment · ~Shane_Gayal1 · 2018-12-29
> > **Modification of Advantage function and Critic in A3C**
> >
> > Can you elaborate more on applying this method with A3C. Especially the modification of the Advantage function?  When calculating the advantage function in A3C , scalar rewards and state-value function is used.   With this method are you proposing to modify only the value function computation in the Advantage function with USF?

---

### Public Comment · ~Shane_Gayal1 · 2018-12-18
**Interesting idea which enables to use USF with large scale RL methods like A3C , IMPALA**

I have noted some interesting facts about the proposed USF architectures by the Authors.  I will point it down since I also wants to know weather I am correct.

1. Previous methods of SF-RL mainly based on DQN architectures and they not highlight any zero shot transfer learning ability.

2. Previous networks have used an autoencoder loss to learn the state representation(Kulkarni et al., Zhan et al.) while training on the DQN task. This can add some instability to the system when the state representation is complex. Barreto et al. (2018) have replaced state representation vectors by assuming the current state as a linear combination of scalar rewards to set of base tasks. With Barreto et al. there should be some base tasks.  Ma et al. have introduced USF idea first where the state representation vector is extracted from an Auto-encoder.  When working with visual states, this can be troublesome. Contrasting to previous work on SF-RL and USF-RL  Authors have used a more straightforward way to predict the state representation with USF which is more scalable.

3. In SF-RL previous work the reward prediction vector also trained by regressing scalar rewards while training on the DQN baseline. In the USF-RL (Ma et al) introduced a goal-oriented reward vector is produced by a neural network that takes the goal as the input. However, still its hard to train the goal-oriented reward vector prediction network due to the sparsity of the reward structure. Because in an example task of navigation, once the training starts agent will see many negative rewards for a long time and decidedly less positive rewards (If its A3C the network will get updated by many agents ).  This makes the training weights of the network unstable. In this paper, authors proposed to use scalar rewards as it is and trained the reward vector prediction network with Q loss.

4.  This combined the general value function approximates(GVFA) with  USF which is useful for the large-scale deep reinforcement learning frameworks like A3C. Because in A3C we can train different agents on different goals. Let's say in a navigation task with different targets, and we can train this whole architecture while learning general patterns.


I also have few questions regarding to train this with complex representations like images for a task of navigation in robotics.


1. Let's say we have a fixed number of targets where each target is sentimentally different from each other, and both the states and targets are represented in images.  So mainly in the architecture proposed by authors, we need to have CNNs for goal embedding, state embedding and reward vector prediction network. How to proceed with this kind of situation ? is it scalable?

2. Do we maintain two CNNs for state embedding network and goal embedding network ?  Cant we use same networks which is more like a Siamese network?  I think this can be a problem when training since practically we only train with given number of goals.

3. Do we maintain two CNN for state embedding network and goal embedding network?  Can't, we use the same networks which is more like a Siamese network?  I think this can be a problem when training since practically we only train with a given number of goals.

4. What is the best way to calculate Advantage in the A3C setting?  To calculate the advantage in A3C, we need a scalar reward at each time steps. If we replace scalar reward as a linear combination of state representations the A3C agent can get unstable because we use the advantage to update the policy directly. So can we use scalar reward as it is to calculate advantage while the USF replaces the value function?

---

### Meta-Review · Area_Chair2 · 2018-12-18
**Reasonable but somewhat incremental result**

**Confidence:** 5
**Recommendation:** Reject

**Metareview:**

In considering the reviews and the author response, I would summarize the evaluation of the paper as following: The main idea in the paper -- to combine goal-conditioning with successor features -- is an interesting direction for research, but is somewhat incremental in light of the prior work in the area. Most of the reviewers generally agreed on this point. While a relatively incremental technical contribution could still result in a successful paper with a thorough empirical analysis and compelling results, the evaluation in the paper is unfortunately not very extensive: the provided tasks are very simple, and the difference from prior methods is not very large. All of the tasks are equivalent to either grid worlds or reaching, which are very simple. Without a deeper technical contribution or a more extensive empirical evaluation, I do not think the paper is ready for publication in ICLR.